# Effect of Process Parameters and High-Temperature Preheating on Residual Stress and Relative Density of Ti6Al4V Processed by Selective Laser Melting

**DOI:** 10.3390/ma12060930

**Published:** 2019-03-20

**Authors:** Martin Malý, Christian Höller, Mateusz Skalon, Benjamin Meier, Daniel Koutný, Rudolf Pichler, Christof Sommitsch, David Paloušek

**Affiliations:** 1Faculty of Mechanical Engineering, Institute of Machine and Industrial Design, Brno University of Technology, Technická 2896/2, 616 69 Brno, Czech Republic; Daniel.Koutny@vut.cz (D.K.); David.Palousek@vut.cz (D.P.); 2Institute of Production Engineering, Graz University of Technology, Inffeldgasse 25F, 8010 Graz, Austria; christian.hoeller@tugraz.at (C.H.); rudolf.pichler@tugraz.at (R.P.); 3Institute of Materials Science, Joining and Forming, Graz University of Technology, Kopernikusgasse 24/I, 8010 Graz, Austria; mateusz.skalon@tugraz.at (M.S.); christof.sommitsch@tugraz.at (C.S.); 4Materials—Institute for Laser and Plasma Technology, Joanneum Research, Leobner Straße 94, 8712 Niklasdorf, Austria; benjamin.meier@joanneum.at

**Keywords:** Selective Laser Melting, Ti6Al4V, residual stress, deformation, preheating, relative density, powder degradation

## Abstract

The aim of this study is to observe the effect of process parameters on residual stresses and relative density of Ti6Al4V samples produced by Selective Laser Melting. The investigated parameters were hatch laser power, hatch laser velocity, border laser velocity, high-temperature preheating and time delay. Residual stresses were evaluated by the bridge curvature method and relative density by the optical method. The effect of the observed process parameters was estimated by the design of experiment and surface response methods. It was found that for an effective residual stress reduction, the high preheating temperature was the most significant parameter. High preheating temperature also increased the relative density but caused changes in the chemical composition of Ti6Al4V unmelted powder. Chemical analysis proved that after one build job with high preheating temperature, oxygen and hydrogen content exceeded the ASTM B348 limits for Grade 5 titanium.

## 1. Introduction

One of the most popular additive manufacturing technologies is Selective Laser Melting (SLM). SLM technology allows the production of nearly full density metallic parts with mechanical properties comparable to the ones produced by conventional methods. Components are made layer-by-layer directly from powdered material, where each layer is selectively melted in an inert atmosphere by a laser beam [1,2,3]. 

Due to non-uniform spot heating and fast cooling, thermal gradients are formed in materials, which lead to the development of residual stresses [1]. Residual stress (RS) is described as stress which remains in the material when the equilibrium with surrounding environment is reached [4]. Unwanted RS in SLM can cause part failure due to distortions, delamination or cracking.

The RS measurement is possible by mechanical and diffraction methods; magnetic and electric techniques; and by the ultrasonic and piezoelectric effect. Mechanical measurements are usually based on material removal and its relaxation or measuring part distortion. Typical mechanical measurement is a drilling method with an accuracy of ±50 MPa [4]. The main advantage of this method is its ability to measure RS to the depth of 1.2 times the diameter of the drilled hole. The Bridge Curvature Method (BCM) is often used in case of the SLM for the fast comparison of process parameters and their influence on the RS [5]. The principle is based on measuring the distortion angle, after the sample is cut off from the base plate, using the bridge-like samples. Measuring accuracy could be affected by imprecise cutting or by angle evaluating method, thus measurement of the top surface inclination was proposed as a better technique [6,7]. Value of the RS can be determined by the simulation of measured distortion in Finite Element Method (FEM) analysis [8].

Ali et al. [9] proved that higher exposure time and lower laser power with preserved energy density lowered the RS due to lower cooling rate and temperature gradient. The variation of laser power and exposure time did not cause a change in yield strength of Ti6Al4V, but elongation increased with lower laser power and higher exposure time. The cooling rate and also the RS can be affected by layer thickness. Higher layer thickness prolonged the cooling rate and RS was lower, but with higher layer thickness the relative density was lowered [9,10].

Scanning strategy has a significant effect on the RS. Ali et al. [11] observed that with longer scanning vectors the RS increased. Due to prolonged time between scanning adjacent scan tracks, higher thermal gradients were induced. The lowest RS had a stripe strategy with the ninety-degree rotation. This conclusion was also confirmed by Robinson et al. [12]. Ali et al. [11] did not observe the positive or negative effect of the scanning strategy on mechanical properties nor relative density. 

Powder bed preheating can significantly reduce the amount of RS [5,13,14]. Ali et al. [15] demonstrated that for Ti6Al4V-ELI material preheating of the build platform to the temperature of 570 °C effectively eliminated the RS. A positive influence of preheating on microstructure and mechanical properties of H13 tool steel was observed by Mertens et al. [16]. The preheating up to 400 °C in his study improved mechanical properties and the parts had more homogeneous microstructure. Formation of cracks can be also affected by preheating, which was observed during printing aluminium of 2618, 7075 [17,18] and tool steel [19].

In this study, the BCM samples made of Ti6Al4V were used to evaluate the effect of process parameters on the relative density and the RS. Investigated parameters were hatch laser speed, hatch laser power border laser velocity, waiting time between adjacent layers and powder bed preheating up to 550 °C. The design of experiment and the surface response method were used for a comprehensive evaluation of the effects of observed process parameters. Furthermore, the influence of high-temperature preheating on powder degradation was evaluated. 

## 2. Materials and Methods

### 2.1. Powder Characterization

In this study, Ti6Al4V gas atomized powder (SLM Solutions Group AG, Lübeck, Germany) was used. The chemical composition of virgin powder delivered by the manufacturer is in Table 1. The powder shape was checked by scanning electron microscopy (SEM) LEO 1450VP (Carl Zeiss AG, Oberkochen, Germany). Figure 1a shows that the powder particles have a spherical shape with a low amount of satellites. The particles size distribution was analysed by laser diffraction analyser LA-960 (Horiba, Kioto, Japan). Measured particle mean size was 43 µm and median size 40.9 µm. The particles up to 29.97 µm represented 10% of particle distribution while particles up to 58.61 µm represented 90% (Figure 1b).

The chemical composition of used powder was evaluated by the following methods. The aluminium content was checked by the inductively coupled plasma atomic emission spectroscopy. Oxygen and nitrogen contents were evaluated by hot extraction in helium by LECO TCH 600 (LECO Corporation, Saint Joseph, MO, USA). The hydrogen concentration was verified by the inert gas fusion thermal conductivity method JUWE H-Mat 2500 (JUWE Laborgeraete GmbH, Viersen, Germany). The accuracy of all methods is Al ±0.327 wt %, O ±0.008 wt % and N ±0.0025 wt %.

### 2.2. Sample Fabrication

The samples were manufactured on the SLM 280^HL^ (SLM Solutions Group AG, Lübeck, Germany) 3D printer. The machine is equipped with 400 W ytterbium fibre laser YLR-400-WC-Y11 (IPG Photonics, Oxford, MS, USA) with a focus diameter of 82 µm and a Gaussian shape power distribution. Argon was used as a protective atmosphere during the process and the O_2_ content was kept below 0.05 %. Before each experiment, the humidity of the powder was measured by the hydro thermometer Hytelog (B + B Thermo-Technik GmbH, Donaueschingen, Germany) with an accuracy of ±2%. The powder humidity was kept under 10%. The heating platform (SLM Solutions Group AG, Lübeck, Germany) was used to preheat the powder. This device is able to preheat the build platform up to 550 °C, but the build area is reduced to a cylindrical shape with 90 mm in diameter and 100 mm in height. For the preheating a resistive heating element is used and the temperature is controlled by a thermocouple placed below the base plate. The temperature of a printed component may be slightly lower than the measured temperature by the thermocouple. However, the maximum height of parts printed in this study is 12 mm, thus the temperature field should be relatively homogeneous. Build data were prepared in Materialise Magics 22.03 (Materialise NV, Leuven, Belgium).

### 2.3. Sample Geometry

The geometry of samples was designed according to the BCM shape (Figure 2a) [5], therefore the effect of chosen process parameters on distortion and RS can be evaluated. Support structures were used for all samples to simulate the condition during the printing of real components. To restrict distortion during the SLM process (before cutting off) the 4 mm high block supports were reinforced with 1 mm block spacing, while fragmentation was switched off. Teeth top length was set to 1 mm. Support structures were added just under the pillars. Samples were rotated to 20° from recoating direction to ensure consistent powder spreading. Samples were cut in the middle of support structures and the evaluated parameter was top surface angle distortion α, which is the sum of α_1_ and α_2_ (Figure 2b).

### 2.4. Design of Experiment

For data evaluation Design of Experiment (DoE) and Surface Response Design (SRD) were used. Hatch laser power (H LP), hatch laser velocity (H LV), border laser velocity (B LV), delay time (DT) and preheating temperature (T) were chosen as the variable factors. The range of parameters with central points is summarized in Table 2. The DT value is waiting delay between two adjacent layers, which was set in the printing machine. The real delay (RD) value which was used for result evaluation is composed of set DT between two layers and 13 s recoating time. If the DT is zero, then the RD value is composed of 13 s recoating time and scanning time. The temperature range was set from the common preheating temperature 200 °C to the maximum temperature of 550 °C that our equipment is capable to evolve.

The half fraction of the SRD was built with the five continuous variable parameters. This means twenty-six samples plus four repetition central points. To minimize the number of global parameters, which has an influence on the whole build job, the face-centered design was used. 

For evaluation, Minitab 17 (Minitab Inc., State College, PA, USA) was used. Data of the top surface angle distortion α were evaluated with full quadratic terms with 95% confidence level for all intervals and with backward elimination of 0.1. Relative density data were evaluated on samples 1–16 as the half fraction of factorial design. Then the data were assessed with 95% confidence level and insignificant term combinations were manually deleted. 

Border laser power was set to 100 W and hatch spacing to 0.12 mm. The layer thickness of 50 µm and stripe strategy with a maximum stripe length of 10 mm and a rotation of 67° was used. Fill contour was turned off. Other parameters were set as standard.

### 2.5. Distortion Evaluation

The 3D optical scanner Atos TripleScan 8M (GOM GmbH, Braunschweig, Germany) was used for assessing distortions of the bridges. Each sample was scanned after cut-off from the base plate and after coating by TiO_2_ mating spray with the thickness of around 3 µm [20]. The 3D scanned surface data were evaluated in GOM Inspect 2018 (GOM GmbH, Braunschweig, Germany). 

The top surface angle distortion α was measured on the top surface of the bridge as is shown in Figure 2b. First, the Computer-aided Design (CAD) data of the undeformed bridge was fitted by Gaussian best fit function on the scanned data. Then three cross sections were created parallel to the YZ plane in distance 0, 8 and −8 mm. Lines using Gaussian best fit function were fitted on the top surface in each cross section (Figure 3a). Next, points in distance 0, 3 and −3 mm in Y direction were created on those three lines (Figure 3b). Then the distance was measured between middle and side points. Finally, X and Z components from each measured distance were used for calculating angle distortions by tangent function. Left and right sides were calculated separately. Therefore, the α value was calculated as the sum of angles on both sides. The result of the top surface angle distortion α value is the mean value of three measurements of one sample.

### 2.6. Relative Density Measurement

Relative density was determined using an optical method and was calculated as the mean value of parallel to build cross sections (Figure 4a). Value of relative density was evaluated in ImageJ v. 1.52k (National Institutes of Health, Bethesda, MD, USA). First, the picture of the cross section was converted to 8-bit type. Next, an automatic threshold was applied and relative density was evaluated in the areas defined by red rectangles (Figure 4b).

## 3. Results

### 3.1. Top Surface Distortion and Relative Density

The experimental design matrix and results of top surface angle distortion α and measured relative density are summarized in Table 3. Samples were sorted in printing order and horizontal lines represent a group of samples which were printed together in one build job.

### 3.2. Surface Response Model for Top Surface Angle Distortion α

Minitab 17 was used to establish a regression model for prediction of the top surface angle distortion α responses to the H LP, H LV, B LV, RD and T. Equation (1) represents the SRM-based mathematical model of significant parameters, which represent the relation between observed parameters. Table 4 and Figure 5 show results from an analysis of variance (ANOVA). The correlation of the regression model for α value is confirmed by determination coefficients R^2^ = 91.82% and adjusted R^2^ = 86.04%.
α = 4.01 − 0.00766 H LP − 0.00854 H LV + 0.003232 B LV + 0.0382 RD − 0.001802 T + 0.000005 H LV∙H LV − 0.000252 RD∙RD + 0.000004 H LP∙H LV + 0.000005 H LP∙T − 0.000002 H LV∙B LV − 0.000018 B LV∙RD − 0.000002 B LV∙T(1)

In order to investigate the effect of high energy and high-temperature preheating on the distortion, an additional four bridge samples were made. Those samples were built with increasing H LP according to Table 5. 

### 3.3. Mathematical Model for Relative Density

Equation (2) represents a mathematical model of significant parameters with an influence on the relative density. Figure 6 shows a Pareto chart of standardized effect for evaluated relative density data. ANOVA results are shown in Figure 7 and Table 6. The correlation of the regression model for relative density value is confirmed by determination coefficients R^2^ = 89.38% and adjusted R^2^ = 82.29%.
Relative density = 136.67 − 0.1558 H LP − 0.0628 H LV − 0.00004 B LV + 0.00766 T + 0.0390 RD + 0.000232 H LP*H LV(2)

### 3.4. Analysis of Used Powder

Figure 8a shows the influence of high-temperature base plate preheating (550 °C) on the powder. The powder significantly changed colour from silver to brown and there is a hint of particle agglomeration. Therefore, the powder used in build job with 550 °C was checked by the SEM microscopy in order to investigate the particle agglomeration and their shape (Figure 8b).

Powder chemistry analysis was done after first build job with preheating to 200 °C and these results were compared with the powder used with 550 °C preheating. Results in Table 7 confirm a rise in oxygen content from 0.12 to 0.33 wt % and in the hydrogen from 0.002 to 0.0168 wt %. Aluminium content also slightly rose from 6.05 to 6.11 wt %, but nitrogen content decreased from 0.017 to 0.0149 wt %. Results are compared with the ASTM B348 Grade 5 titanium requirements and virgin Ti6Al4V powder chemical composition received by the vendor.

## 4. Discussion

### 4.1. Top Surface Angle Distortion α

The main contribution of each parameter on the distortion, and therefore the amount of residual stresses, can be derived from the ANOVA (Table 4). Further on the significance of each parameter can be evaluated by a p-value. If the value is lower than 0.05, then the parameter is significant, while the p-value of the lack-of-fit parameter should be high, which shows that the error value is not significant. The p-value of the lack-of-fit parameter 0.685 shows that the regression model for the top surface angle distortion α fits the measured data. 

Table 4 and Figure 5 show that the most significant parameters for reduction distortions are T and H LP with 46.31% and 17.22% linear contribution. P-values of those parameters are 0. The RD and H LV are not that significant in comparison with the previous two parameters. Their linear contributions are 5.26% and 3.62%, while p-values are lower than 0.05, therefore parameters are significant. The B LV can be considered as an insignificant parameter with 0.59% linear contribution and the p-value greater than 0.05. Figure 5 indicates that H LP, B LV and T have linear behaviour in contrast to RD and H LV.

From the ANOVA results, it can be concluded that increasing the preheating temperature or laser power causes a reduction in deformation. In contrast, increasing H LV, B LV and RD lead to higher deformations. This can be contributed to the cooling rate, which is lower with slower laser movement, higher preheating and shorter waiting time. Therefore, the thermal gradients, residual stresses and finally distortions are lower [9,11,15,21,22]. 

Optimal parameters for achieving the lowest distortion can be predicted from the fitted regression model (Figure 9). For reaching the lowest distortion it is predicted to use H LP 275 W, H LV 785 mm/s, B LV 350 mm/s, RD 17 s and T 550 °C. Predicted α is −0.182°. In contrast, the lowest distortion predicted with a preheating temperature of 200 °C is 0.139°.

Figure 10 shows the influence of energy density (ED) on the top surface angle distortion α. Samples used for this comparison were made with the same preheating temperature of 550 °C and 200 °C. The effect of B LV and RD was neglected. The interpolated line for 200 °C samples is constantly dropping with increasing ED and as was measured by Mishurova et al. [7], and this trend constantly continues. In contrast, the interpolated line for 550 °C samples with added high ED samples starts much lower than 200 °C line. The decrease of α value is gradual until 65 J/mm^3^ and drops rapidly with higher ED. 

### 4.2. Relative Density

The most significant effect out of involved parameters influencing the relative density are H LP and H LV. The linear contribution is 34.66% for H LP and 23.85% for H LV. They are also significant in their two-way interaction with a contribution of 26.41%. P-values are 0 for H LP and 0.002 for H LV. 

Preheating temperature has a linear contribution of 2.88% while its p-value of 0.153 is higher than 0.05. This means that this parameter in the observed range is not that significant for the model due to the high contribution of H LP and H LV. Relative density rose with higher T. Positive influence of preheating was also confirmed on stainless steel M2. It was proved by Kempen et al. [19] that with higher preheating temperature, higher laser velocities can be used while maintaining relative density.

The real delay has a linear contribution of 1.58% and a p-value of 0.277, which means the RD has minimal influence on relative density. Prolonged RD leads to an increase in relative density. 

Border laser velocity with a linear contribution of 0 % and p-value 0.993 means that this parameter was not significant for the model, which can be due to the place where porosity was measured.

From the ANOVA (Figure 7) it can be deduced that a sample will have maximum relative density when values of H LP, RD and T are set as the highest and H LV as the lowest. This means the highest energy density is in the hatch. 

### 4.3. Powder Degradation

There is clear evidence that the chemical composition of the powder significantly changed due to oxidation. Titanium alloys suffer high chemical affinity to oxygen leading to form a thin oxide layer even on air room temperature. Exposing titanium to an oxygen-containing atmosphere at elevated temperatures around 550 °C increase diffusion rates through thin oxide layers, and allows penetration of oxygen in the material [23]. After experiments with 550 °C preheating, oxygen content increased against 200 °C preheating from 0.12 to 0.33 wt % which is 0.13% higher than the ASTM B348 requirement for Grade 5 titanium. 

Increased oxygen content in the Ti6Al4V causes an increase in yield and ultimate tensile strength, whilst ductility up to 0.19 wt % of the oxygen content remains constant [24]. Ti6Al4V additively manufactured alloy is due to rapid cooling mainly composed from α’martensitic microstructure even with preheating up to 550 °C [15]. Therefore, this is sensitive to oxygen content because of the concentration of oxygen higher than 0.22 wt % leads to the brittleness of the α’martensitic structure. Critical oxygen content for α and β structure is 0.4 wt % [25]. Oxygen concentration above 0.25 wt % leads to change in the typical microstructure, which causes a sharp decrease in ductility of Ti6Al4V [26].

Hydrogen content in the powder used under 550 °C exceeds the approved ASTM B348 limit value of 0.0125 wt % and its content rose from 0.002 to 0.0168 wt %. The diffusion rate of the hydrogen is rapidly increasing at the elevated temperatures [23]. The origin of the hydrogen element is most probably from powder moisture which was kept below 10%. Hydrogen in titanium alloys causes a phenomenon known as hydrogen embrittlement and could lead to part failure [27,28].

It was shown that the critical issues with processing the titanium alloy by SLM at high temperatures are connected with chemical composition changes in the unused powder, although the material was processed under argon atmosphere with oxygen concentration of 0.05% and powder humidity kept below 10%. The measured concentration of oxygen and hydrogen was beyond ASTM B348 requirement for Grade 5 titanium. Therefore, the used powder cannot be used for mechanical stressed parts.

## 5. Conclusions

Effects of hatch laser power, hatch laser velocity, border laser velocity, preheating temperature and delay time on residual stress and relative density on SLM processed Ti6Al4V samples have been investigated. In addition, the impact of preheating temperature up to 550 °C on Ti6Al4V powder degradation has been discussed. The main findings are the following:-The preheating temperature has the main effect on the distortion and residual stress out of all observed parameters. With a high preheating temperature of 550 °C, the distortions of the top surface decreased and the relative density increased. The linear contribution effect of preheating was 46.31% on the distortion and 2.88% on the relative density.-Relative density mainly depends on the hatch laser power and hatch laser velocity.-Higher energy density decreased the deformations of the BCM samples. The value of the top surface distortion α decreased from 0.363° to 0.098° with increased energy density from 65.5 to 83.3 J∙mm^−3^.-Longer delay time negatively influenced distortions, but improved relative density. The linear contribution effect of the delay time was 5.26% on the distortions and 1.58% on the relative density.-Powder bed preheating to 550 °C led to fast powder degradation. The oxygen and hydrogen content rose beyond the ASTM B348 requirement for Grade 5 titanium after one build job.

## Figures and Tables

**Figure 1 materials-12-00930-f001:**
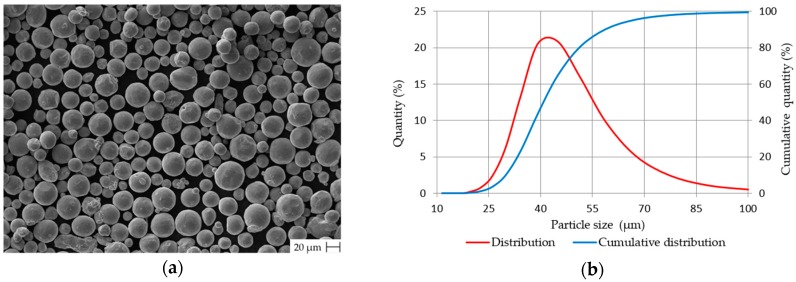
Ti6Al4V gas atomized powder characterization (**a**) shape evaluation by SEM; (**b**) particles size distribution.

**Figure 2 materials-12-00930-f002:**
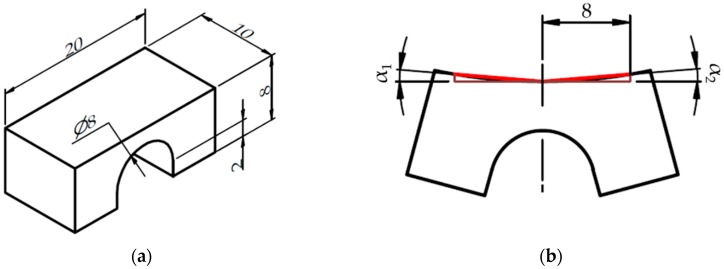
Samples geometry: (**a**) Dimensions of the BCM sample; (**b**) Measured bridge top surface angle distortion α is the sum of α_1_ and α_2_. Dimensions presented in mm.

**Figure 3 materials-12-00930-f003:**
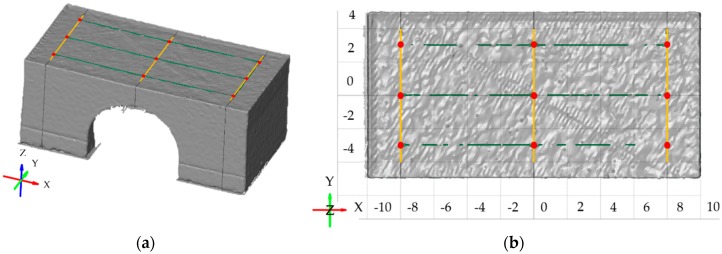
Distortion evaluation, fitted lines are yellow, line points are red and measured distances are green: (**a**) Isometric view on scanned data; (**b**) Top view of scanned data.

**Figure 4 materials-12-00930-f004:**
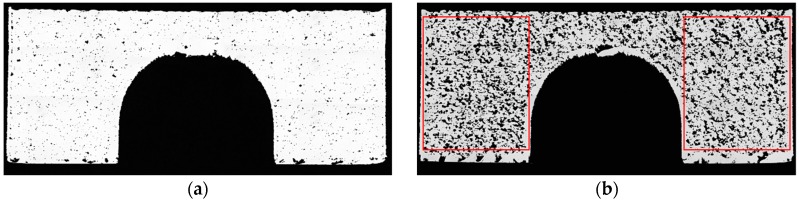
Cross sections of the BCM samples: (**a**) Cross section of sample 1; (**b**) Cross section of the sample made with the lowest energy density (Sample 3), red rectangles show area for relative density evaluation.

**Figure 5 materials-12-00930-f005:**
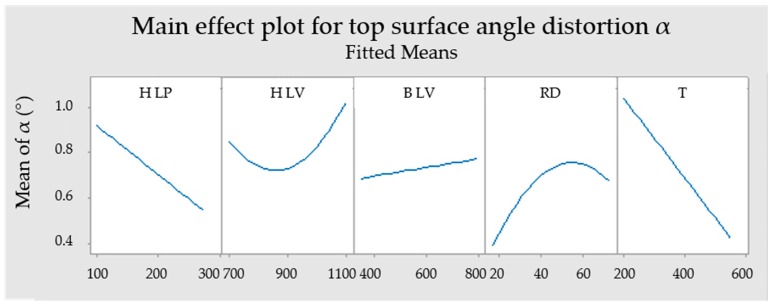
Main effect plot for top surface angle distortion α.

**Figure 6 materials-12-00930-f006:**
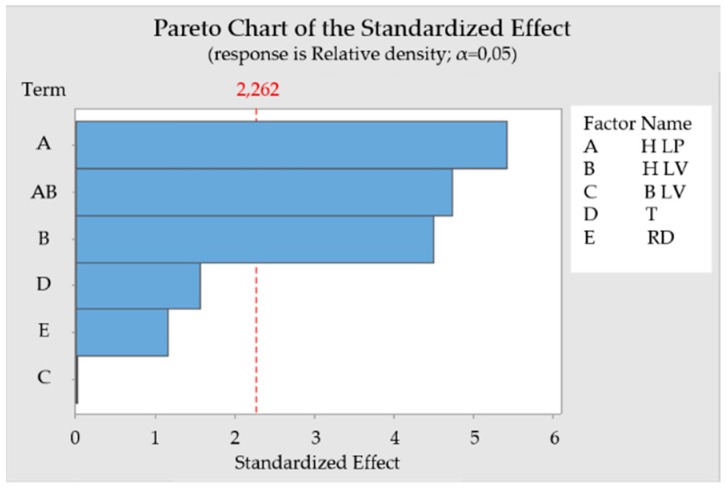
Pareto chart of the standardized effect to relative density.

**Figure 7 materials-12-00930-f007:**
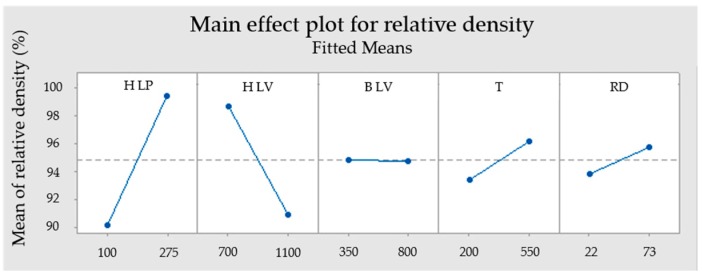
Main effect plot for relative density.

**Figure 8 materials-12-00930-f008:**
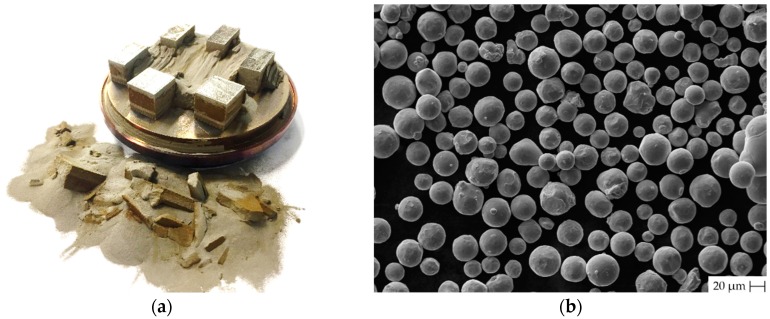
The powder used in heating unit preheated to the 550 °C (**a**) Build job made with 550 °C; (**b**) SEM microscopy photo of the powder used with 550 °C.

**Figure 9 materials-12-00930-f009:**
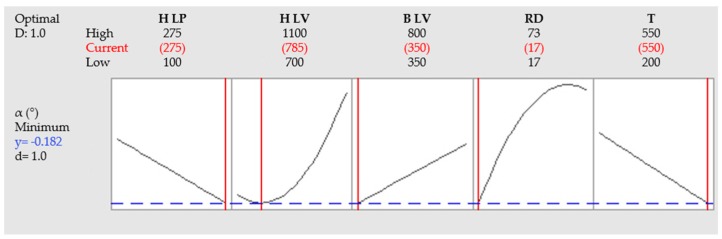
Predicted values for the lowest distortion in the full range of observed parameters.

**Figure 10 materials-12-00930-f010:**
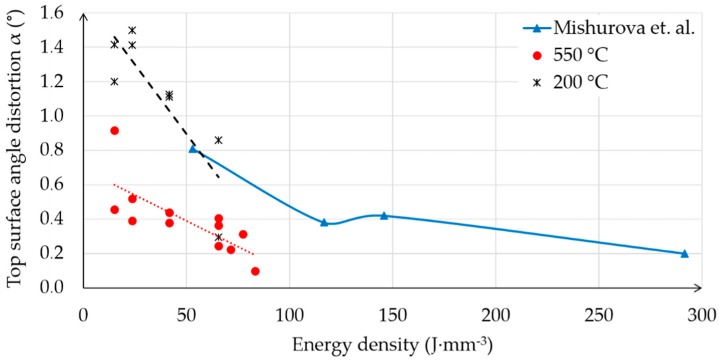
Effect of energy density on the top surface angle distortion α.

**Table 1 materials-12-00930-t001:** Chemical composition of virgin Ti6Al4V powder.

Al (wt %)	C (wt %)	Fe (wt %)	V (wt %)	O (wt %)	N (wt %)	H (wt %)	Ti (wt %)
6.38	0.006	0.161	3.96	0.087	0.008	0.002	Bal.

**Table 2 materials-12-00930-t002:** Table of used process parameters for Design of Experiment (DoE) and Surface Response Design (SRD).

Values/Parameters	H LP (W)	H LV (m∙s^−1^)	B LV (m∙s^−1^)	DT (s)	T (°C)
Minimum value	100	700	350	0	200
Middle point	187.5	900	575	30	375
Maximum value	275	1100	800	60	550

**Table 3 materials-12-00930-t003:** DoE and SRD test matrix with process parameters, the value of top surface angle distortion α and relative density.

Sample Number	H LP (W)	H LV (mm/s)	B LV (mm/s)	TD (s)	RD (s)	T (°C)	α (°)	Relative Density (%)
1	100	700	800	0	22	200	1.499	97.59
2	275	700	350	0	22	200	0.294	98.68
3	100	1100	350	0	22	200	1.201	74.04
4	275	1100	800	0	22	200	1.110	99.97
5	275	1100	350	60	73	200	1.127	99.60
6	100	700	350	60	73	200	1.413	97.06
7	100	1100	800	60	73	200	1.416	81.33
8	275	700	800	60	73	200	0.859	99.37
9	275	1100	350	0	22	550	0.437	99.69
10	100	700	350	0	22	550	0.389	98.68
11	275	700	800	0	22	550	0.406	99.43
12	100	1100	800	0	22	550	0.456	82.33
13	100	1100	350	60	73	550	0.917	91.17
14	100	700	800	60	73	550	0.520	98.93
15	275	1100	800	60	73	550	0.377	99.35
16	275	700	350	60	73	550	0.244	99.51
17	100	900	575	30	43	375	0.905	-
18	187.5	1100	575	30	43	375	0.943	-
19	187.5	900	575	30	43	375	1.000	-
20	187.5	900	800	30	43	375	0.771	-
21	275	900	575	30	43	375	0.454	-
22	187.5	900	350	30	43	375	0.764	-
23	187.5	700	575	30	43	375	0.795	-
24	187.5	900	575	30	43	375	0.740	-
25	187.5	900	575	0	17	375	0.174	-
26	187.5	900	575	60	73	375	0.876	-
27	187.5	900	575	30	43	550	0.392	-
28	187.5	900	575	30	43	200	0.809	-
29	187.5	900	575	30	43	375	0.716	-
30	187.5	900	575	30	43	375	0.668	-

**Table 4 materials-12-00930-t004:** ANOVA table for the top surface angle distortion α.

Source	DF	Contribution (%)	Adj SS	Adj MS	F-Value	P-Value
Model	12	91.82	3.44257	0.28688	15.9	0
**Linear**	5	73.01	2.81581	0.56316	31.21	0
H LP	1	17.22	0.64558	0.64558	35.78	0
H LV	1	3.62	0.13584	0.13584	7.53	0.014
B LV	1	0.59	0.03329	0.03329	1.85	0.192
RD	1	5.26	0.26462	0.26462	14.66	0.001
T	1	46.31	1.73648	1.73648	96.23	0
**Square**	2	4.73	0.17728	0.08864	4.91	0.021
H LV∙H LV	1	1.79	0.17109	0.17109	9.48	0.007
RD∙RD	1	2.94	0.11008	0.11008	6.10	0.024
**2-Way Interaction**	5	14.08	0.52797	0.10559	5.85	0.003
H LP∙H LV	1	1.95	0.07309	0.07309	4.05	0.060
H LP∙T	1	2.90	0.1089	0.10890	6.04	0.025
H LV∙B LV	1	2.68	0.10043	0.10043	5.57	0.031
B LV∙RD	1	4.62	0.17310	0.17310	9.59	0.007
B LV∙T	1	1.93	0.07244	0.07244	4.01	0.061
**Error**	17	8.18	0.30677	0.01805	-	-
Lack-of-Fit	14	6.40	0.24014	0.01715	0.77	0.685
Pure Error	3	1.78	0.06662	0.02221	-	-
**Total**	29	100.00	-	-	-	-

**Table 5 materials-12-00930-t005:** Parameters of samples with increasing laser power and value of α.

Sample Number	H LP (W)	H LV (mm/s)	B LV (mm/s)	RD (s)	T (°C)	H Ed (J∙mm^−3^) ^1^	α (°)
31	275	700	350	22	550	65.5	0.363
32	300	700	350	22	550	71.4	0.224
33	325	700	350	22	550	77.4	0.313
34	350	700	350	22	550	83.3	0.098

^1^ Calculated as H Ed = H LP∙(H LV∙Lt∙Hs)^−1^, Layer thickness (Lt) = 50 μm, Hatch spacing (Hs) = 120 μm.

**Table 6 materials-12-00930-t006:** ANOVA table for relative density.

Source	DF	Contribution (%)	Adj SS	Adj MS	F-Value	P-Value
Model	6	89.38	893.804	148.967	12.62	0.001
**Linear**	5	62.96	629.660	125.932	10.67	0.001
H LP	1	34.66	346.611	346.611	29.36	0
H LV	1	23.85	238.471	238.471	20.20	0.002
B LV	1	0	0.0010	0.001	0	0.993
T	1	2.88	28.756	28.756	2.44	0.153
RD	1	1.58	15.821	15.821	1.34	0.277
**2-Way Interaction**	1	26.41	264.144	264.144	22.38	0.001
H LP∙H LV	1	26.41	264.144	264.144	22.38	0.001
**Error**	9	10.62	106.239	11.804	-	-
**Total**	15	100.00	-	-	-	-

**Table 7 materials-12-00930-t007:** Chemical composition analysis of the Ti6Al4V powder.

Powder State/Checked Elements	Al (wt %)	O (wt %)	N (wt %)	H (wt %)
ASTM B348 Grade 5	5.50–6.75	Max. 0.20	Max. 0.050	Max. 0.0125
Virgin Ti6Al4V	6.38	0.087	0.0080	0.0020
Ti6Al4V 200 °C	6.05	0.120	0.0170	0.0020
Ti6Al4V 550 °C	6.11	0.330	0.0149	0.0168

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
