# Peer review of "Effect of Process Parameters and High-Temperature Preheating on Residual Stress and Relative Density of Ti6Al4V Processed by Selective Laser Melting"

_materials, 2019, doi:10.3390/ma12060930_

Round 1

Reviewer 1 Report

The Selective Laser Melting process includes a number of process variables which makes the process complex. Using a DoE study as in the present manuscript is a good way to explore in a controlled way the influence from varying several of these process parameters on properties of built material (residual stresses in this case). I find the research design and methods to be mostly well described, but one thing that I miss is an explanation for selecting 550C as the elevated temperature? You refer to some reference where 570C was tried and that this temperature reduced residual stresses, but why did you select 550C in your study? After working in Aerospace industry for the past 20 years with titanium alloy applications, we are well aware of the oxidation phenomena that occurs with titanium (pure Ti as well as with Ti Alloys) when exposed to temperatures exceeding >480C. Above this temperature the oxygen diffusion rate increases dramatically. Oxygen as such is one of the most important alloying elements in Ti Alloys for increased strength, but as you mention in the manuscript the maximum content for most Ti-64 alloy material forms is 0.18-0.20wt% O. When exceeding this amount of oxygen the ductility degrades and the risk for formation of detrimental alpha case increases (oxygen enriched areas which are very brittle). Taking this into consideration, and the results that you present here that the oxygen take up is so much that it exceeds the maximum allowed oxygen content for this alloy in the powder that was pre-heated to 550C, means that the powder has to be scraped after the first build. I.e. it would not be possible to re-use any of this Ti-64 powder since the oxygen (and also the hydrogen content!!) exceeds the maximum allowed concentration. Adding virgin powder to this high-oxygen containing powder to get an “average” oxygen concentration below maximum allowed oxygen content could perhaps be considered once or twice, but since local high oxygen content is enough to form hard and brittle alpha case (which is detrimental to fatigue properties) this would not be acceptable for aerospace applications. Below I add a few references on this. And since the powder is so expensive, if non-melted powder cannot be re-used/recycled I am convinced that this would never become economically viable for manufacturing of any real components. On one hand then I ask myself what interest do these results have for the end users that want to be able to manufacture successfully Ti-64 parts with the SLM process? Very little I would say. On the other hand, if you would not have done this study, part of your findings might not have come to common knowledge, which is arguments for why these results are important to publish. You refer to a reference [15] where they did a similar experiment but used a pre-heating at 570C, maybe that study showed the same results regarding oxygen take up etc? I have no time to go through that now unfortunately.

I am still puzzled to why you selected the pre-heating temperature of 550C? You do not anywhere explain why? With the knowledge of the oxidation kinetics of Ti and its Alloys at temperatures >480C when exposed to oxygen containing atmospheres, I personally would have selected a pre-heating temperature around 450-480C instead of 550C. Then the risk of oxygen take-up would be much less and maybe the residual stresses formed in the built materials would still be relatively small compared with the materials built at lower temperature?

Another thing that I also would like to know is how you measured the temperature in the chamber during building? Is there only a thermocouple placed on the build plate, or are there other thermocouples also installed in the chamber to measure the temperature further up in the build? I mean there is likely to be a temperature difference from the solid build plate as compared to higher up in the powder build, right? You mention nothing about this uncertainty, and that needs to be added in the text of the manuscript. I believe that others have investigated this before, so maybe at least you can refer to others who did investigate temperature variations in the chamber during building?

To summarize, I find this manuscript to include some interesting results that would fit well to publish in this journal and the special issue, but before recommending it for publication I request the authors to reply to the following:

1)    Explain and include in the manuscript why you chose 550C as pre-heating temperature for the elevated build experiments.

2)    Explain/discuss the temperature measurement of the chamber temperature; how did you measure it and what uncertainties exists with this way of measuring it?

3)    On line 208 in the manuscript change to “Table 6”, not “Table 4”.

Author Response

Dear Reviewer,

Authors would like to thank you for your constructive recommendations and remarks. We have included all corrections into the new version of our manuscript. You can also find some minor language corrections in the new manuscript version. The answers to your questions are the following:

Point 1: Explain and include in the manuscript why you chose 550°C as pre-heating temperature for the elevated build experiments.

Response 1: Temperature range between 200 and 550°C was chosen because 550°C is the maximum temperature which is our device capable to preheat the powder bed and 200 °C is common preheating temperature. We wanted to observe the temperature range above common temperatures up to our maximum temperature. Now with Surface Response Model, we can predict the behaviour of the distortions in the full range of our preheating capability. Even for the lower temperatures which are more suitable for the Ti6Al4V powder because of powder degradation.

Ali et al. [15] have chosen the temperature of 570 °C    because it is 30°C below the minimum temperature for the martensite decomposition. Which was not focus of this study. Ali et al. preheated the powder even up to 770°C, but they did not mention the chemical change of the powder. To my best knowledge, I do not know about any other publication where they check the influence of high-temperature preheating on powder degradation.

In chapter 2.4. (line 131-133) has been added the sentence:

The temperature range between 200°C and 550°C was set from the common preheating temperature to the maximum temperature that our equipment is capable to evolve.

Point 2: Explain/discuss the temperature measurement of the chamber temperature; how did you measure it and what uncertainties exist with this way of measuring it?

Response 2:  For the preheating a resistive heating element was used. The temperature was controlled by a thermocouple placed below the base plate. Since the powdered material has much lower thermal conductivity than solid material the temperature may vary in the height of powder on the base plate. However the printed part is connected directly to the base plate so the temperature can spread throw part which has a high thermal conductivity, but still, the temperature of a printed component may be slightly lower than the measured temperature by the thermocouple. However, the maximum height of parts printed in this study is 12 mm thus the temperature field should be relatively homogeneous.

            Infrared heaters can be used for homogeneous preheating of the powder which some other researchers developed for that purposes. All the others are using resistive heating element placed under the base plate and their equipment is mostly developed by them self.

In chapter 2.2 (line 106-109) has been added the description of the temperature measuring system and the uncertainties were discussed.

Point 3: On line 208 in the manuscript change to “Table 6”, not “Table 4”.

Response 3: The numbering of all tables were changed, cause we have added one extra table in chapter 2.1.

Reviewer 2 Report

Ref: materials-413622

Title: Effect of Process Parameters and High Temperature 2 Preheating on Residual Stress and Relative Density 3 of Ti6Al4V Processed by Selective Laser Melting

Journal: Materials

This paper presents the effects of hatch laser power, hatch laser velocity, border laser velocity, preheating temperature and delay time on residual stress and relative density on Ti6Al4V samples produced by SLM. Experimental and numerical discussions were conducted. Results indicate that high preheating temperature has the most effect on reduction the residual stress but in the same time increased the relative density and caused changes in the chemical composition of the Ti6Al4V powder.

There are a number of details, presented below, that require further attention by the authors.

Comments and suggestions

1. Introduction

Page 1, line 36: insert “.” (full stop) after citation in text of references

2. Powder Analysis

a. Specify the O, Al and H content of the Ti6Al4V powder before preheating, as received from the vendor (within the section 3.4. Used Powder Analysis).

b. Explain in brief the possible causes that have led to changing the chemical composition of the powder under the effect of preheating temperature (within the section 4.4. Powder Degradation).

3. References

Page 12: Please, revise references according to requirements from Instructions for Authors.

Author Response

Dear Reviewer,

Authors would like to thank you for your constructive recommendations and remarks. We have included all corrections into the new version of our manuscript. You can also find some minor language corrections in the text in the new manuscript. The answers to your questions are the following:

Point 1: Page 1, line 36: insert “.” (full stop) after citation in the text of references.

Response 1: The citation on line 36 has been corrected according to correct citation style.

Point 2: Powder Analysis

a.       Specify the O, Al and H content of the Ti6Al4V powder before preheating, as received from the vendor (within section 3.4. Used Powder Analysis).

            Response 2a: The powder chemical specification delivered by the vendor has been                        added in chapter 2.1. (line 85) and in Table 7 in section 3.4 (line 228).

b.       Explain in brief the possible causes that have led to changing the chemical composition of the powder under the effect of preheating temperature (within section 4.4. Powder Degradation).

             Response 2b: Short explanation was added in section 4.4. on line 288-292 and 303-312.

Point 3: References. Page 12: Please, revise references according to requirements from Instructions for Authors.

Response 3: References were corrected according to Materials citation style.